# “It Almost Makes Her Human”: How Female Animal Guardians Construct Experiences of Cat and Dog Empathy

**DOI:** 10.3390/ani12233434

**Published:** 2022-12-06

**Authors:** Karen M. Hiestand, Karen McComb, Robin Banerjee

**Affiliations:** 1Mammal Communication and Cognition Research, School of Psychology, University of Sussex, Brighton BN1 9QH, UK; 2School of Psychology, University of Sussex, Brighton BN1 9QH, UK

**Keywords:** animal empathy, IPA, anthropomorphism, anthropocentrism, human-animal interaction

## Abstract

**Simple Summary:**

As highly social animals, humans experience better mental and physical health when they have access to social support. One aspect of social support is the extent to which those in supporting roles empathise with us. Many humans use–and rely upon–companion animals to provide social support and believe them to be empathic in times of need. This study used in depth interviews with six participants who identified as having had experiences when their dog or cat empathised with them, to examine how they made sense of these experiences. The participants consistently reported that changes to their animal’s normal behaviour was key to identifying animal empathy, but there was significant variation in their understanding of how and why their animals performed empathic actions. Inconsistencies in participant explanations may illustrate the difficulties in understanding animals’ emotions, motivations, and cognitive abilities in light of a history of denial of animal capacities on one hand, coupled with burgeoning scientific evidence about animal communication on the other. The findings in this study can be applied in areas where companion animals are used explicitly for social support, such as animal-assisted therapy and emotional support animals.

**Abstract:**

Understanding how humans perceive and construct experiences of non-human animal empathy (hereafter, ‘animal/s’) can provide important information to aid our understanding of how companion animals contribute to social support. This study investigates the phenomenology of animal empathy by examining how humans construct sense-making narratives of these experiences, with the hypothesis that anthropomorphic attributions would play a key role in these constructions. Comprehensive, semi-structured interviews were conducted with six participants, using established interpretative phenomenological analysis methodology to facilitate deep examination of how they interpreted and reacted emotionally. Participants were consistent in reporting changes to their companion animals’ normal behaviour as the key to the identification of animal empathy experiences, yet they were highly paradoxical in their constructions of perceived internal drivers within their dogs and cats. Explanations were highly dichotomous, from highly anthropomorphic to highly anthropocentric, and these extremes were combined both within individual participant narratives and within some thematic constructs. This research demonstrates that experiences of companion animal empathy can be powerful and meaningful for humans, but the inconsistent mixture of anthropomorphic and anthropocentric reasoning illustrates the confused nature of human understanding of animals’ internal states. Insight into how humans construct animal empathy has implications for the moral status of these animals and an application for companion animals used explicitly for social support, such as in animal-assisted therapy and emotional support animals.

## 1. Introduction

As highly social animals, humans experience better mental and physical health and cope with stressors better when they have access to social support [1]. One antecedent of social support is the extent to which those in supporting roles empathise with us [2]. The term ‘empathy’ is often used ambiguously in the scientific literature, but for the purposes of this work we take empathy to involve emotional, cognitive, and behaviourally expressive aspects, and to entail an observer perceiving another’s affect and experiencing shared feeling [3]. 

Companion relationships with non-human animals (hereafter ‘animal/s’) have evolved over 15,000 and perhaps as long as 40,000 years [4]. They are reported to be positive for our mental and physical health [5,6]. This phenomenon is known as the ‘pet effect’ [7]. While there is debate as to the veracity of the positive effect of companion animals due to contrasting results [8], studies that consider attachment and social support theories suggest that non-human animals fulfil human needs for emotional support [9], even acting as substitutes for reduced human support networks [10]. However, the role of animal empathy towards humans in generating this social support has not been explicitly investigated. 

Attributing uniquely human capacities to non-human entities is considered anthropomorphic [11]. Despite a human tendency to anthropomorphize literally anything [12], the primary target remains animals [13]. While examination of the phenomenon of anthropomorphism is accelerating [14,15], being anthropomorphic is often considered unscientific and viewed negatively by the scientific community (see [16]), though anthropomorphism can have positive impacts on human–animal relationships. As a counterpoint, anthropocentrism has been defined as the interpretation of reality according to human values, needs, and experience, due to a belief structure where humans are primary amongst all species [11]. We can perceive our companion animals and their capacities through either lens, either affording them human-like capacities, perhaps beyond their physiological and cognitive abilities, or denying them such affordances based on a bias that views humans as exceptional. 

While there is mounting evidence for canine empathic abilities [17], the study of feline empathy lags far behind–for example, a major review of emotional contagion research in mammals included no references to studies in cats [18]. Regardless, owners retain beliefs that both cats and dogs can empathise with us [19]. 

As mutual caring, reciprocal support, and empathy moderate human relationships, it is possible that these same attributes play a role in the bond humans have with their companions. Hence, examining how we perceive and construct animal empathy experiences can generate valid and important information to aid our understanding of how animals provide social support–in particular, by revealing the extent to which people use anthropomorphic explanations for experiences of empathy from canine and feline companions.

This study investigates the phenomenology of animal empathy by focussing on how humans construct sense-making narratives of animal empathy experiences. We hypothesized that anthropomorphic attributions would play a key role in these constructions. To elucidate a deep understanding of each participant’s experience and draw interpretative meaning from them, a qualitative approach concerned with subjective experience, and in particular emotional responses, is essential. Therefore, the current study used the qualitative methodology of interpretative phenomenological analysis to gain insight into how participants identified and constructed a lived experience of animal empathy.

## 2. Materials and Methods

### 2.1. Theoretical Framework 

As the research question centred on how participants identified and understood their experiences of animal empathy, this study utilized the qualitative approach of interpretative phenomenological analysis (IPA), which focusses on participants’ lived experiences and the meaning they make of them. This method facilitates the deep examination of experiential phenomena and is particularly beneficial for understanding how participants interpret and react emotionally to the experiences of interest. IPA is an inductive method and is the product of a joint elucidatory process in which, not only does the participant interpret their lived experience, but the analyst ultimately provides their account of what they think the participant is thinking, resulting in a ‘double hermeneutic’ [20] (p. 80). 

### 2.2. Participants 

Upon ethical approval (ER/KH447/1, University of Sussex), the participant sample was generated purposefully via social media advertising (Facebook) and word of mouth. Eligibility criteria were deliberately wide to promote participation, meaning any adult (over 18 years) who self-identified as having lived experience of an occasion when they believed their companion animal was empathic towards them was included. All participants who came forward were female, and experiences discussed were evenly split between dogs and cats. Two participants were residents of New Zealand, the remaining four in the United Kingdom, and as such all were English speaking and derived from broadly similar western cultural backgrounds. This study followed recommendations for IPA methodology to be applied to a sample size of one to six participants [20] (p. 51). Participants gave voluntary verbal consent before interviews took place.

### 2.3. Interviews 

Semi-structured interviews were conducted by a single interviewer (author 1) following the established IPA methodology [20] (chpt. 4). This paper addresses themes arising from part B, questions 4, 5, and 6 of the interview schedule (Table 1). Themes arising from parts A and C are interpreted in future work.

The interview schedule and interview technique were piloted with two unanalysed participants, after which suitable amendments were made. Each interview lasted approximately one hour, and all were conducted online via a video conferencing platform suitable for non-sensitive data (Zoom) between March and May 2021. Recordings were immediately downloaded to a secure university server and then deleted from the online platform. Interviews were recorded and transcribed automatically by the video conferencing platform, with transcriptions later checked against audio recording and manually corrected by the interviewer to ensure verbatim accuracy. 

### 2.4. Data Analysis 

Interviews were analysed sequentially by the interviewer, and recruitment terminated when themes reached saturation. Transcripts were first read, and re-read, to ensure familiarity with content, then exploratory comments were made line by line, which were categorised as descriptive, linguistic, or conceptual [20]. An interpretation was then conducted by the systematic coding of transcripts using proprietary software (NVIVO release 1.5) followed by clustering of evolving themes. Emerging themes were examined for divergence, convergence, repetition, and nuance, and this process was repeated for each transcript to uphold a commitment to each participant’s meaning-making. Reflexivity was enhanced by cycling back over previously analysed materials in an inductive cycle to move the interpretation from the individual level to a gestalt understanding of relationships between themes. Generated themes and coded data were discussed in tandem with psychological knowledge from other authors throughout the analysis to test and develop the plausibility and coherence of the interpretative account. 

## 3. Results

Two superordinate themes were identified. The first covered the context and identification of animal empathy experiences, while the second encompassed multiple themes and sub-themes concerned with how participants constructed their experiences. Sub-themes with interpretive commentary and illustrative extracts are presented below, with those concerning how animal empathy is constructed by guardians further interpreted through anthropomorphic, mixed, or anthropocentric lenses (Figure 1). Participants are anonymised and pseudonyms are used for animal names.

### 3.1. Context and Identification of Animal Empathy

Participants reported a variety of contexts where self-identified experiences of animal empathy took place. Some described empathic interactions in terms of entirely emotional support such as in situations of grief, loneliness, and stress, others in terms of physical support including protection and illness, while several participants described both emotional and physical support. In this extract, Participant E describes a period of grief after the death of a close friend and the emotional support role of Barney (dog) during that time:


*PE: But sometimes there’ll be something [ ] and it takes my breath away, and it’s almost like sometimes that he (Barney) picks up on that and will just come and lean on me or will come and flop next to me or something. And I do find it really comforting.*


In all cases participants identified a change from their cat or dog’s normal behaviour as the indicator of an empathic interaction. In the following extract Participant F describes the actions of Henry (cat) during a period of convalescence:


*PF: Me and Henry would always lie [ ] we had a particular position that we always lay in, and I was on my back and she sat right on my womb, where I’d had this horrendous operation, and just sat there. So, it was not a position she’d normally sit in at all and if I tried to move, she’d hiss at me and she never hissed at me either...*


In this extract the narrator uses alteration in behaviour to identify that Henry is attempting to care for them bodily. This extract also shows that in common with all interviews, Participant F used their animal’s increasing physical proximity as an identifier of an empathic interaction, as does Participant C when speaking about Tukker (dog):


*PC: he’d seek you out and try to initiate contact if he could see you’d had a crap day, you know, come and put his head on my lap, he’d come wriggle up to me.*


In this extract Participant C also attributes Tukker with the ability to identify (‘he could see’) their emotional need. This is illustrative of the following thematic framework which attempts to unpick the diversity of how animal empathy experiences are constructed and understood by participants. 

### 3.2. Constructions of Animal Empathy

How participants understood what was going on inside their animal during their experience of animal empathy varied across a spectrum from highly anthropomorphic to highly anthropocentric, with some explanations involving a mix of both (Figure 1). Multiple explanations were used by each participant, with some participants expressing conflicting constructs within their reasoning. 

#### 3.2.1. Anthropomorphic Constructions

Many participants provided explanations that utilised human-like capacities to construct understanding of their experiences of animal empathy.

##### Cognitive Attribution

The most anthropomorphic explanations provided by participants were those attributing high levels of cogitation and intention by the animals involved, such as shown in this extract from Participant A:


*PA: Bay was thinking ‘mum’s in trouble’ or ‘mum’s getting hurt and I need to do something about it’, [ ] like ‘I have to protect mum’ [ ]*


Here the participant ascribes Bay (dog) not only with understanding of the context of what was happening (an incident of domestic violence) but also of conscious thought and action intention. By giving Bay an internal ‘voice’, this participant also assumes that Bay categorizes their relationship in a familial way and sees Participant A as ‘mum’.

Participant F likewise affords Henry (cat) with the ability to apply human-like cognition as they recovered from painful abdominal surgery: 


*PF; it was exactly where it was hurting me, absolutely, and it’s like she completely knew, and she was just like, ‘just lie the fuck down, keep still you’re not well, I want you to recover’. And she sort of looked after me all through the next week, when I was in recovery from the operation.*


Here Participant F apportions conscious knowing to Henry, including what was wrong (pain, not well) and what needed to happen (lie down, keep still), and as with the previous extract, also ascribes Henry an internal ‘voice’. 

##### Exceptionalism

Several participants expressed beliefs that their animal companion’s exceptionality explained how they were able to empathize with humans. Participant B describes the exceptional abilities of their cat as even allowing them to transcend species:


*PB: she’s very, yeah, very unique [ ] I think that just makes her, just almost makes her human, though she’s not human obviously, (inaudible), but it almost makes her slightly human in what she does so I think she is very special. [ ] because she does these things like that humans would do, and I think that’s probably how I feel about her, more than other cats we’ve had because they just acted like normal cats.*


Here the anthropomorphic classification is made explicit alongside the elevation of this cat from others of its species. Several participants spoke of their animals as ‘unique’, as a descriptor of their identity and in terms of their capacities being an extension of what a ‘normal’ animal could do. Similarly, Participant G singled out one cat in their household for the ability to pre-empt and warn them of oncoming seizures. 


*PG: I think she’s highly intelligent, and has managed, because she’s highly intelligent to understand what her normal senses are telling her.*


This participant understood this cat’s ability to predict seizures in terms of its exceptional intelligence, particularly in comparison to other cats, and indeed humans. This speaks to a view that for animals to understand our internal states and to communicate this to us requires skills beyond the capability of their conspecifics, sometimes conferring on them a human-like status. Against a backdrop of social norms that views many empathic and cognitive capacities as exclusively human, to explain how these animals have acted in these experiences, guardians may feel they have to separate their animal from the norm. 

#### 3.2.2. Mixed Constructions

Some constructions mixed anthropocentric and anthropomorphic interpretations as in the following sub-themes. 

##### Special Senses

Most participants utilized some degree of folk rationale to explain their animal’s behaviour. These explanations centred around beliefs of animal knowing and the attribution of special, non-human senses: 


*PB: Whether animals have got another, an extra sense we don’t know [ ] I don’t know what it is that they feel or can sense but there’s obviously something. Because they seem to try and be more of a comfort to you for that little period of time [ ] you think ‘why are they doing this,’ but I think they must have some sort of sense that they, that you need help.*


Explaining animal empathy this way suggests that the animal’s actions were so inexplicable by any other means that abilities unknown or unknowable to humans must be at play. This suggests that participants were sometimes reticent to attribute human empathic capacities, perhaps due to concerns over allegations of anthropomorphism.

However, the attribution of special senses was sometimes considered superior to human empathy:


*PF: you can trust them to sort of know, and maybe have some kind of superior knowledge in certain situations, like okay, that’s what she thinks, that’s what should happen [ ] so I felt that cats did do that you know, they were capable of sort of targeted comfort, like knowing when you need something.*


Participant F emphasized their trust in Henry (cat) and based this on a belief that Henry had access to knowledge that that wasn’t available to humans. Considering special senses from this perspective could also be construed as highly anthropomorphic, in that Henry is in possession of ‘more than human’ capacities.

##### Surprise/Expectation

Participants often expressed surprise at the empathic actions of their animal. 


*PF: it was amazing, because yeah, she was a cat! [ ] You’re like wow! [ ] yeah, I was really surprised [ ]. But yeah, it was really, I really was amazed, I was really like wow, Henry you know you’re doing here.*


Here, Participant F’s surprise can be interpreted from as anthropocentric, in that non-human capacities are generally expected to be inferior, hence any display of capacity beyond an accepted animal norm is worthy of astonishment.

While the previous extract illustrates wonder at animal knowing, other participants were more anthropomorphic and held expectations of their animal companions:


*PA: so no, I knew in the moment what it was, and I wasn’t surprised, like I wasn’t surprised at all yeah [ ]. No, no, not at all, and I feel like he’d do it again. He, if it happened again with someone else I can hundred percent guarantee he would do it again yeah.*



*PB: Because they seem to try and be more of a comfort to you for that little period of time, which is quite, you know, you think ‘why are they doing this’ [ ] And I just grew up thinking all cats could do that, but then people tell me, no, no, no.*


This final extract illustrates the contradiction within this theme by simultaneous expressing surprise; questioning why the animal is providing comfort, followed by an expectation that this is just something cats can do. It also illustrates a tension some participants expressed attributable to believing in their animals’ capacities while in an anthropocentric culture as discussed in the next theme.

#### 3.2.3. Anthropocentric Constructions

Three sub-themes are interpreted as displaying commonality in anthropocentricity. These sub-themes illustrate a belief structure that sees humans as primary amongst species and the participant explanations are generated through a human-centric lens. 

##### Proofs

All participants repeatedly stated proofs to verify their attributions of empathy. This usually took the form of detailed and persistent explanations of the identifier of animal empathy–behaviour change. 


*PA: I got up to just use the bathroom that night, and usually Bay couldn’t care less, he’ll just keep sleeping, but this night he actually got up and came to the bathroom, which is very unusual. [ ] Bay just slept there with his head on my chest. He would usually start the night sleeping on my chest anyway, but he gets really hot and then he goes to my feet. That night, he was just like on my chest, the whole night.*


The deliberate caregiving described can be interpreted against a human-centric cultural backdrop whereby to make assertions of animal empathic capacities requires extensive and robust proof to protect from accusation of naive anthropomorphism. Participants were thus motivated to provide multiple proofs to show that their animal’s behaviour was not merely chance or being misinterpreted. 

##### Physiological Explanation

Some participants employed concepts of normal physiological functioning to construct their experiences. These explanations used the sensitivity of animal senses (smell, hearing, etc.) to interpret human internal states. These rationales were rooted in reality as opposed to more magical abilities discussed in the special senses theme. 


*PA: [ ] even down to my heart rate or how my physiology is changing and Bay is just sensing it better than a human would sense it. Like, he can probably tell my heart rate’s gone up, that ever so slight decibel of my voice has gone up, he can probably like, smell it from my, like, hormones coming off me because I’m stressed, [ ] I feel like they know what we’re thinking, [ ] like a physiological way, and then they react accordingly,*



*PG: Scientists say that the smell, is picking up some kind of smell I give out, that I can’t smell, but because she’s got far more receptors than a dog has [ ] that she can pick it up easier.*


Participant A had an educational background that informed their more detailed physiological explanation, while Participant G relied on folk knowledge, but both remain based in the reality of existing senses and physical functioning. Using physiological constructs to understand animal empathy behaviours is a more parsimonious explanation than those used in anthropomorphic or mixed reasonings and shows how the participants were at times wary of over-attributing empathic capacities to their non-human companions. 

##### Black Box 

The most anthropocentric explanations are grouped into a theme labelled black box in reference to a historical view of non-human animals as simple, stimulus-response organisms with no or limited conscious intention to their actions. This view has its roots in Cartesian thought and became validated for a time through the work of Skinner and the behaviourist tradition. 


*PE: I think that’s a higher-level thinking than I imagine Barney could have. [ ] There’s no thought behind it, it’s just a spontaneous emotional reaction, when a dog is happy and he wags his tail, if he’s nervous his tail goes down, if he’s cross he barks, if he’s frightened he growls [ ] So I think he can recognize your emotions in that very basic way, but I don’t think my emotions would have an impact on how he was feeling.*


Here, Participant E also rejects the possibility of emotional contagion (the emotional state matching of one individual to another [18] from humans to animals, as does Participant A: 


*PA: dogs and cats in general, are not going to feel sad just because I’m feeling sad. I think they’ll react to it, that is just my thinking [ ] but I don’t think that just because their human is sad that they’re just going to get sad, like I don’t I don’t think that’s how they would work.*


Both participants, when musing on the internal workings of their companion animals, describe the empathic behaviours as spontaneous or reactionary, in essence, incognizant. This interpretation follows the behaviourist tradition which afforded no internal awareness to non-human animals. 

While these extracts demonstrate a most parsimonious and anthropocentric interpretation, the very same participants also expressed highly anthropomorphic readings of the same experiences (e.g., Participant A extract, cognitive attributions theme, where they describe Bay (dog) as consciously thinking that ‘mum’s in trouble’). Similarly, despite Participant E describing Barney (dog) as being unable to have higher level thinking, they did attribute some cognitive abilities:


*PE: I do feel that he thinks he’s looking after me. That he’s, in that moment, keeping me, I don’t mean safe physically, but just keeping me okay [ ] he’s acknowledging that perhaps there’s something wrong and his closeness is perhaps a comfort. yeah.*



*Interviewer: Do you think he’s choosing to do that, like he’s making a choice to fulfil that function?*



*PE: yeah, without a doubt, I think he recognizes it and he, yeah chooses or decides to just come and sit with me at that, at that point.*



*Interviewer: Do you think he knows your sad?*



*PE: Yes, I don’t know, why but yes, I think he does.*


While demographic data of participants were not expressly obtained in this study, it was apparent during the interviews that Participant A and E had educational backgrounds that included psychology, which may have informed their reticence to express unfettered anthropomorphic explanations to their experiences, resulting in their representation in opposing thematic constructs. 

## 4. Discussion

Participants were consistent in reporting changes to their animal’s normal behaviour as key to the identification of animal empathy experiences, yet they were highly paradoxical in their constructions of the internal drivers within the animal. Dichotomous explanations ranged from highly anthropomorphic, where animal companions knew what their humans were thinking, feeling, and needed, to highly anthropocentric expressions of animals as little more than stimulus-response organisms. Furthermore, there was a combination of these extremes both within individual participant narratives, and within some thematic constructs. The narratives also conformed to the social support theory of human-animal relationships. Devoldre et al. [2] describe two positive forms of social support, emotional and instrumental. Emotional support is that which assists the management of emotions, which all participants narratives contained, whereas instrumental support is characterised by more problem-orientated help. Participant A, F, and G all described specific physical instrumental support provided by both cats and dogs. 

In contrast to the investigation of accuracy and functionality, there has been relatively limited exploration of the psychological basis of anthropomorphism [21], and while debate continues as to the accuracy versus erroneousness of anthropomorphic attributions in companion animals, that anthropomorphism is an intrinsic aspect of human nature is less controversial. Anthropomorphic thinking varies between people [22], and previous work has shown it to be a stable trait in individuals [23]. However, the findings of the current study suggest that variability can also exist within individuals, with seemingly incompatible views being held simultaneously. This finding may relate to evidence in developmental psychology where a body of work shows that learners hold misconceptions about phenomena based on naïve theories gained from observation of the environment during their lives and go on to use multiple and sometimes contradictory explanations based on superficial reasoning to explain an event. Furthermore, acknowledging contradictions is avoided by modifying observations to defend previously held views [24].

The range of how participants constructed and understood animal empathy experiences may represent an inherent confusion as to what is really happening within their animals during perceived empathic encounters. Epley, Waytz, and Cacioppo [14] put forward a model of anthropomorphism that combines both motivational and cognitive aspects, and this provides a framework to account for and predict this variably. This model proposes three psychological factors: accessibility and applicability of knowledge about humans (elicited agent knowledge), motivation to explain and understand the behaviour of non-humans (effectance motivation), and the desire for social contact (sociality motivation). 

The elicited agent knowledge factor affords that the accessibility of knowledge about us as humans plays a central role in attributions to non-humans. As we have such immediate access to rich phenomenological information of what it is like to be ourselves, this forms a rapid and automatic basis for applying that knowledge to non-human agents. That anthropomorphic explanations featured in all our participant narratives conforms to this factor. Furthermore, this factor suggests that when internal knowledge is less accessible, it is less likely to be applied. This aspect may be seen in our data where some participants extended beyond anthropomorphic descriptions and into a more-than-human realm of magical thinking, ascribing special senses to their animal companions. Perhaps, if the internal psychological mechanisms of empathy are difficult or inaccessible knowledge for some, it then becomes difficult to apply to animals. This may be an explanation for the resulting attribution of magical capacities to explain the unintelligible. 

Motivational factors in the model provide modulation to the degree of anthropomorphism used by participants. Sociality motivation relates to the desire to establish social connections and predicts that attributions to animals are increased in the absence of connections to other humans [14]. In the context of social support and empathy, this may be particularly relevant, as evidenced by several participants describing the context of their empathy experiences as times of loneliness and loss of close human companions. A motivation to reduce discomfort associated with uncertainty over the actions of non-humans, and improve the prediction of future behaviour by providing anthropomorphic explanations for animal actions is termed effectance motivation. As Nagal [25] would have it, we cannot know what it is like to be a bat, or indeed our companion cats and dogs, hence there is a motivation to interpret their behaviour rather than leave it unexplained. In this study, there is the added incentive to explain the animals’ behaviour anthropomorphically because to do so increases the emotional support provided by the encounters if the animals are believed to be empathic. 

After rapid application of elicited agent knowledge to provide anthropomorphic explanations, the model suggests there is post hoc correction to accommodate evidential knowledge of non-human capacities. Participants who reported expertise in psychology and science appeared to conform to this aspect of the model as they were more careful to provide highly parsimonious explanations for their experiences, perhaps due to a greater understanding of the negative view of anthropomorphism as folk or naive reasoning. However, it was also these participants that displayed the most notable dichotomy in their narratives, perhaps illustrating a greater cognitive dissonance between the internal motivation to anthropomorphise and cognitive desire to correct in light of their knowledge. As anthropomorphism is driven by both motivational and cognitive determinants, the mixing of interpretations both thematically and in individual participants may represent the various ways participants combined and rationalised these competing methods of constructing their experiences. 

The three-factor model of anthropomorphism assists us in understanding some aspects of the participant narratives, but how might we understand the more anthropocentric themes? De-mentalisation is a strategy unconsciously used by people to alleviate cognitive dissonance experienced by what is known as the ‘meat paradox’–the inconsistency of loving some and animals and eating others [26]. For example, humans tend to deny food animals the capacity to suffer more so than they do for companion species [27]. Perhaps providing anthropocentric explanations, particularly those that deny the animals’ emotional repercussions or contagion from their owner’s distress, is motivated by similarly extending deniability of their capacity to be negatively affected, thus assuaging any guilt owners may feel for using their animals for social support. 

An important theme uncovered in this study was that of exceptionalism. That some participants viewed their animals as exceptional in comparison to others shares commonality with the concept of subtyping of stereotypes in human prejudice literature. Subtyping refers to the separation of members of a stereotyped group into a separate category because they violate rules of the stereotype [28]. As the exceptionalism theme emerged in these data, it suggests that stereotypes about dog and cat empathic abilities exist and, as display of animal empathy was a violation, the stereotype is likely to lean toward the anthropocentric side of the spectrum. 

### Limitations

In comparison to other research methodologies, the sample size used in this work may appear both small and biased. However, the purpose of this work is not to provide statistical or population-level generalisability; instead, in approaching the research question via an IPA method, our aim was to achieve theoretical generalisability and provide novel insights into the topic, which may then be taken forward via other research methods [29]. 

When using IPA, it is appropriate to purposively recruit a sample that is relatively homogeneous regarding the topic of interest, and due to level of detail required in analysis of phenomenology, small sample sizes are advised [20] (p. 49). This resulted in participants that were not only homogeneous with regard to their experiences, but also their gender, language (English), and western cultural background. These aspects must be taken into account when considering the insights generated in this work. In particular, as the construction of human–animal relationships is a semiotic process whereby the meanings generated from signals and experiences are hugely influenced by the culture of the participants, it would be interesting to investigate constructions of animal empathy experiences in participants of different cultural backgrounds. 

Researcher experience with qualitative interviewing techniques can impact the quality of such work, and while the lead researcher was new to this method, thorough piloting of the interview schedule, strict adherence to substantiated IPA data analysis protocols, and clear establishment of rapport with participants leaves the authors confident that the resulting richness of the interview data provides relevant and compelling findings. A further recognised limitation is that some participants may have been reticent to express anthropomorphic views to a research scientist, perhaps skewing those participants towards more anthropocentric or parsimonious explanations. In future work it is suggested that greater demographic detail is gathered, such as the timescale of animal relationship, participant education level and background, and perhaps to consider blinding participants to the interviewer’s scientific background.

## 5. Conclusions

Themes identified in this study provide valuable and rich insight into how humans understand their companion animals. This research demonstrates that experiences of companion animal empathy can be powerful and meaningful for humans, but the inconsistent mixture of anthropomorphic and anthropocentric reasoning illustrates the confused nature of human understanding of animals’ internal states. As increasing public knowledge of the burgeoning scientific evidence of animal capacities intersects with a long history of anthropodenial [30] and aspersion of anthropomorphism, this confused state may not quickly dissipate. However, as the ascribing of internal states–particularly emotions–to animals has important implications for their moral status [31], gaining understanding and insight into how humans construct animal empathy may hold an applied value. For example, this knowledge could lead to more targeted education in areas where humans use companion animals for social support, such as in animal-assisted therapy and emotional support animals. 

## Figures and Tables

**Figure 1 animals-12-03434-f001:**
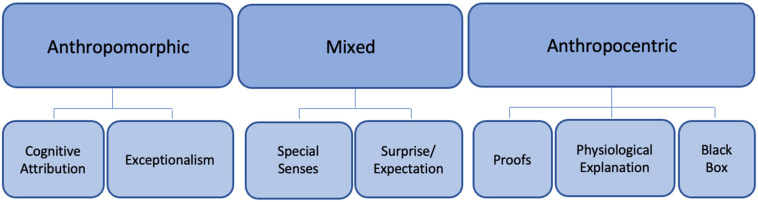
Constructions of Animal Empathy.

**Table 1 animals-12-03434-t001:** Interview Schedule.

(A) Relationship before
1. How would you describe your relationship with animals in general?
2. Can you tell me briefly about your history of pet ownership
3. Can you tell me a brief history of your relationship with *name of dog or cat*
(B) Experience of companion animal empathy
4. Tell me about the event/situation when you felt *name of dog or cat* helped you physically or emotionally, cared about you, consoled or supported you, or you thought seemed to understand your emotions?
5. What do you think *dog or cat* was thinking or feeling during this interaction?
6. How did you did you feel when *name of dog or cat* acted this way?
7. Can you describe how you felt about *name of dog or cat* after this event?
8. Did this experience change how you think about other animals?
(C) General beliefs about empathy and role in HA relationships
9. Do you think cats and dogs can feel empathy towards humans?
10. What does the term empathy mean to you?
11. How did this experience impact on you and/or *name of dog or cat*? Were there any consequences?
12. Is there anything you would like to add that this discussion has bought up for you?

The italicised words are replaced by the actual name of animal during the interview.

## Data Availability

The data presented in this study are available on request from the corresponding author.

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
