# Peer review of "“It Almost Makes Her Human”: How Female Animal Guardians Construct Experiences of Cat and Dog Empathy"

_animals, 2022, doi:10.3390/ani12233434_

Round 1
Reviewer 1 Report
“It almost makes her human”: How animal guardians construct experiences of cat and dog empathy
Overall comments
Thank for you the opportunity to review this paper. I found this a very interesting read and an important topic to explore given the increased awareness and application of animal sentience.
I can see application of this topic much more widely and its value in enhancing our awareness of animals and their communicative abilities is important also.
Study premise and background is justified and study design and approach clear and appropriate.
The paper is very well written and constructed overall.
Simple Summary
No comment – clear, concise, precise.
Abstract
Clear and concise. Appropriate for study. Highlights key areas
Line 23 – ‘these dogs and cats’ feels jarring as no prior comment about them – could clarity be enhanced by changing to companion animals?
Introduction
Clear with good definitions and clarity of background of study.
Overall, introduction is good with nice consideration of current state of play in subject area.
Well written and to my knowledge, incorporates relevant key studies and info that is pertinent to this work.
Line 60 – general comment here but worth considering – should ‘animal/animals’ be replaced throughout by ‘non-human animals’ to avoid confusion and a level of speciesism? There is a move in anthrozoology to say ‘non-human animals’ instead of just animals to avoid anthropocentrism. However, in the context of this sentence, animals might be ideal.
Materials and Methods
Clear and well-structured for clarity. Focus on IPA and lived experience is valid and robust for this study.
Ethical review noted.
Interesting all participants were female. No other demographic information is detailed – this *might* be relevant for later results and discussion, especially because of the focus on lived experience – this could have impacts on the study outcomes and conclusions drawn, because of intersectionality?
Data analysis appears consistent and appropriate for study with evidentiary background
Clear methodology overall – nothing specific to note beyond points above
Results
Line 146 – how were the ‘lenses’ determined – at what point was it mixed or one or the other? – Ah – I apologise – picked up from line 174 – could this be clarified earlier for enhanced flow?
Line 348 – see earlier point about demographic info and possible impact of this on outcomes – you might not have this however and that is understandable.
Overall – the description of qualitative data is clear and relevant to the study – on a purely personal note, I might like more breakdown of participant response categorisation for clarity and understanding of context, but I also understand why this is not here.
Discussion
General – see earlier comment about animal vs non-human animal
Relevant and consistent discussion of findings.
Line 365-377 – query whether there is value in discussing or at least mention value of ‘good’ anthropomorphism for animal welfare/wellbeing? Relevant for study impact.
Line 396 – Could this statement be clarified or reworded or even evidenced? ‘As the psychological mechanisms of empathy may be relatively inaccessible knowledge to some people’
Limitations are acknowledged – the point about participant reticence is interesting – how could this be managed? Is there an evidence base of supporting this going forward as this seems to be a significant limitation in the application of the study outcomes – the link between cognitive and affective behaviour……thinking and doing.
Conclusion
Clear. Good review and reflection and awareness of relevance of study outcomes
References
I have not exhaustively gone through these, but all appear fine – present and correct
Author Response
We wish to extend our gratitude to Reviewer One for taking the time to conduct a thoughtful and considered review of this paper. We have amended the manuscript as suggested, itemised edits are included below:
- Line 23: changed 'these dogs and cats' to 'companion animals' as suggested
- Line 60: suggestion to use 'non-human animal'. Thank you for raising this, it is an issue we wrestled with as we agree that this is the more appropriate term. However, for readability, we have used 'animal' throughout the paper, now with the addition on Line 23 (abstract) and Line 50 (introduction) of non-human animal (hereafter 'animal/s') to raise the point.
- Line 146: added reference to Fig 1. to clarify lenses earlier in the results
- Line 348: regarding demographics and impact on results. Details on demographic characteristics were not obtained in this study and this is reflected in the amended limitation section (Line 472). The methods (Line 102-110) have also been amended to include a little more detail on criteria for recruitment.
- The limitations section has been revised (Line 449-475) to reflect the point that understanding of the findings ought to include consideration of the language, culture and gender of the participants.
- Line 365-377 good anthropomorphism: we have added Line 64-65 to make this excellent point.
- Line 396-402: amended to make this point clearer.
- Limitation of participant reticence: this is another excellent point, and we have added a little on Lines 472-475 to suggest ways to manage this.
Thank you again for your very helpful review.
Reviewer 2 Report
I would like to thank the authors for examining the important, yet challenging topic of animal empathy as interpreted by animal guardians. The themes described in this study help to further knowledge on how anthropomorphism and anthropocentric thinking factors into explanations of animal empathy experiences. However, as noted in your discussion, a few participants had educational backgrounds that influenced some of their responses. I recommend some more discussion of how demographic characteristics such gender, age, etc. may also influence the participants interpretations. How long did each person have their companion animal? As this information is not reported, it is difficult to know whether these factors could be influencing their responses. Due the sample size, it is also difficult to generalize these results. I think increasing the sample and including more analysis of demographic characteristics would help strengthen these findings.
Author Response
Many thanks to Reviewer Two for taking the time to conduct a thoughtful and considered review of this paper. We have amended the manuscript as suggested, itemised edits are included below:
- Details on demographic characteristics were not expressly obtained in this study and this is reflected in the amended limitation section (Line 472). the methods (Line 102-110) have also been amended to include a little more detail on criteria for recruitment.
- With regards to the ability to generalised the findings of the work, please see expanded limitations section (Line 449-475) which seeks to address some of your concerns on this point. Most pertinently we have now included a reference to an excellent paper by Varipio et al (2021) that provides helpful discussion on differing scientific perspectives (post-positivist/objectivisit vs. constructivist) and how best to consider a work's standard of rigorous in light of the methodological approach. These authors make the argument that in constructivist qualitative work, such as this paper, it is unhelpful to apply post-positivist measures of rigorous (such as concerns over biased samples or generalisability to a wider population). Instead, they suggest that the rigorous of such work is predicated on its success in theoretical generalisability - the extent to which the insights gained are able to develop or map onto existing theory (in this case Epley, Waytz and Cacioppo's three factor model of anthropomorphism) not to generalise to wider populations.
Thank you again for your considered and helpful review.
Reviewer 3 Report
Could the authors please supply full addresses? Centres and schools lie within larger organisations/addresses
I am aware that phenomenological work often has a small sample, but the work still has to be viewed through this lens. As such, how the information can be extrapolated is constrained by the sample.
As all participants were female, the themes etc. within the work can really only be applied to the female lived-experience (the title should reflect this). The work doesn't expand on cultural origins but it does suggest these are English-speaking individuals, possibly with European heritage. This brings with it a substantial historical bias from the anglosphere where considerations of "animals as automata" have greater leverage thanks to European male philosophers. Again, it should be emphasised that this is not a "human-level" effect. This needs to be considered throughout the paper in terms of its wider applicability but also its limitations.
As part of the anonymisation of the data and excerpts I would encourage the authors to remove the animals' names. It may seem rather unlikely, but it could be possible for people to identify the participant through their pet's names/characteristics.
Be careful to check the quotes (e.g. 189 "mum's in trouble/getting hurt")
Please provide more information on the participants where possible, especially around cultural background.
Author Response
We wish to extent our thanks to Reviewer Three for taking the time to conduct a thoughtful and considered review of this paper. We have amended the manuscript as suggested, itemised edits are included below:
- Full addresses are supplied as suggested, thank you for noticing this omission.
- Female only sample: title amended as suggested and Lines 104-111 have been amended to include a little more detail on criteria for recruitment.
- Extrapolation of the work from sample: please see the expanded limitations section (Line 449-475) which seeks to address some of your concerns on this point. Most pertinently we have now included a reference to an excellent paper by Varipio et al., (2021) that provides helpful discussion on differing scientific perspectives (post-positivist/objectivist vs. constructivist) and how best to consider a work's standard of rigor in light of methodological approach. These authors make the argument that in constructivist qualitative work, such as this paper, it is unhelpful to apply post-positivist measure of rigor (such as concerns over biased samples or generalisability to a wider population). Instead, they suggest that the rigor of such work is predicated on its success in theoretical generalisability - the extent to which the insights gained are able to develop or map onto existing theory (in this case Epley, Waytz and Cacioppo's three factor model of anthropomorphism) and not to generalise to wider populations.
- Cultural recognition of participants: thank you for raising this important point, please see Line 456-464 for addition to the manuscript to reflect this.
- Animal names: thank you for bringing this up, it was an issue we wrestled with during the write up as we felt it was important to include animal names to ensure due respect was paid to the non-humans included in the work. As such, their names have been changed to pseudonyms to protect anonymity.
- Line 193 (previously 189) quote: grammatical issues in quote amended
Thank you again for your very helpful review.
Reviewer 4 Report
Comments on the reviewed manuscript “ It almost makes her human: How animal guardians construct experiences of cat and dog empathy” submitted to the Animals
General comments
I appreciate the opportunity to review this manuscript.
The manuscript reports the attribution of empathy in pets (dogs and cats) of six English-speaking women. The qualitative method used by the authors is the IPA. The result generated a discussion of how anthropomorphism and anthropocentrism are shaping the tutors' perception of their pets' empathy.
The manuscript is scientifically based and original. I believe that there is a theoretical and empirical contribution to the understanding of empathy in dogs and cats, although it still generates controversies that can be discussed in future studies.
In the introduction it is not well defined what "empathy" is. There are some definitions of animal empathy for humans, but none are clearly defined in the manuscript.
The method has a consistent theoretical basis, but I have some doubts about the generalization of the study because the sample included only female, English-speaking tutors. The criteria for recruiting the interviewees was also not clear, which concerns whether there is a bias in the sampling.
The animals, and the relationship with the tutors, is a semiotic construct, with meaning and signifiers, which in turn are the result of people's cultural structure. Authors can consult the scientific literature on the symbolic construct of what dogs and cats are in each culture (Indian, Chinese, Japanese, Persian, Latin American, etc.). Therefore, the interpretation of the present study cannot neglect this aspect.
I believe that the authors should better explain how the interviewees were recruited and how to avoid sampling bias. The discussion needs to highlight cultural differences. I suggest that in the limitations of the study, the sample restriction of only English-speaking women should be noted.
Author Response
Many thanks to Reviewer Four for taking the time to conduct a thoughtful and considered review of this paper. We have amended the manuscript as suggested and itemised edits are included below:
- Definition of empathy: given the variety of, and debate over, empathy definitions we had avoided explicitly stating a definition other than to include where there is consensus that it involves emotional, cognitive and behavioural aspects. However, we have added information to Lines 47-50 to clarify our position. As this paper does not present data on the meaning that participants attach to the term empathy, we felt it didn't not illuminate the findings to delve further into the convoluted scientific debate associated with defining it.
- Criteria for recruitment: please see Lines 104-111 that have been amended to include a little more detail on this point.
- Generalisability and possible bias: please see expanded limitations section (Line 449-475) which seeks to address some of your concerns on this point. Most pertinently we have now included a reference to an excellent paper by Varipio et al., (2021) that provides helpful discussion on differing scientific perspectives (post-positivist/objectivist vs. constructivist) and how best to consider a work's standard of rigor in light of the methodological approach. These authors make the argument that in constructivist qualitative work, such as this paper, it is unhelpful to apply post-postitivist measures of rigor (such as concerns over bias in samples or generalisability to wider populations). Instead, they suggest that the rigor of such work is predicated on its success in theoretical generalisability - the extent to which the insights gained are able to develop or map on to existing theory (in this case Epley, Waytz and Cacioppo's three factor model of anthropomorphism) and not to generalise to wider populations.
- Notwithstanding the prior point, the question raised regarding the effect of culture on the semiotic construct of the human animal relationships explored in the paper was an excellent one that we thank the reviewer for highlighting. Please see Line 459-463 for addition to the manuscript to reflect this.
Thank you again for your considered and helpful review.
Round 2
Reviewer 2 Report
I would like to thank the authors for making the suggested revisions, especially in addressing the limitations of the study, and I think the manuscript has improved. I understand the justification for the inclusion of a small sample, but wonder if it could still help to make this apparent in the summary, abstract, and possibly the title to be more transparent about the implications of the findings.
For line 13 of the summary: change to "interviews with 6 participants..."
For line 30 in the abstract: change to "The six participants were consistent in reporting..."
Author Response
Thanks again to reviewer 2 for their time. Great suggestion regarding highlighting the number of participants, this has been done as suggested in the summary (Line 13) and in abstract (Line 28).
Reviewer 3 Report
Thanks for your clarifications and amendments which place the work in a clearer context in terms of extrapolation and generalisability.
Author Response
Thanks again for your time.